# Combination Treatment of Locoregionally Aggressive Granulomatosis with Polyangiitis and Cranial Base Infiltration

**DOI:** 10.3390/brainsci13081140

**Published:** 2023-07-29

**Authors:** Krzysztof Bonek, Eliza Brożek-Mądry, Jakub Wroński, Mateusz Płaza, Agnieszka Zielińska, Katarzyna Helon, Krzysztof Wójcik, Małgorzata Wisłowska

**Affiliations:** 1Department of Rheumatology, National Institute of Geriatrics, Rheumatology and Rehabilitation, 02-637 Warsaw, Poland; 2Department of Otorhinolaryngology, National Institute of Medicine of the Ministry of Interior and Administration, 02-507 Warsaw, Poland; 3Department of Radiology, National Institute of Geriatrics, Rheumatology and Rehabilitation, 02-637 Warsaw, Poland; 42nd Department of Internal Medicine, Jagiellonian University Medical College, 31-007 Krakow, Poland; 5POLVAS Consortium, Jagiellonian University Medical College, 31-007 Krakow, Poland

**Keywords:** granulomatosis with polyangiitis, temporal bone infiltration, skull base, cerebellopontine angle

## Abstract

Objectives: To present a personalized approach in three cases of treatment-resistant, locoregionally aggressive forms of cANCA-positive granulomatosis with polyangiitis (GPA) and skull base involvement. Methods: Three patients with GPA and skull base involvement were described alongside a critical review of the current literature. Results: All presented patients suffered from GPA with an inflammatory tumor at the skull base, alongside cerebellopontine angle involvement, cranial nerve palsies, cerebellar disorders, concomitant hearing loss, and severe otalgia. Symptoms were associated with progressive granulomatous destruction of the temporal bone, laryngopharynx, and central nervous system infiltration. Treatment with cyclophosphamide and high doses of glucocorticoid steroids were ineffective but subsequent therapy with rituximab was successful in the presented cases. The literature review showed that the course of the disease with skull base involvement is associated with poorer clinical and radiological responses to standard pharmacotherapies. Conclusion: Granulomatous inflammation localized in the skull base is associated with a more aggressive disease progression and is less likely to respond to pharmacotherapy. Standard induction therapy with cyclophosphamide and glucocorticoid steroids may be ineffective. A better response may be achieved by using rituximab and concomitant local treatment with glucocorticoid steroid injections.

## 1. Introduction

Granulomatosis with polyangiitis (GPA) is a chronic systemic necrotizing granulomatous vasculitis, which is classified as a small vessel inflammation associated with antineutrophil cytoplasmic antibodies (ANCAs) that target proteinase 3 (PR3 or cANCA) [1]. GPA is characterized by multiple organ involvement, including lungs, kidneys, and the ear, nose, and throat region (ENT) [1]. Neurological manifestations of GPA are less common, with peripheral nervous system involvement occurring in just 23% of GPA cases, compared to 17–75% of all ANCA-associated vasculitis (AAV) [2]. Central nervous system (CNS) involvement occurs even less commonly, with only 3–6% of patients having granulomas within their skull bones [3].

A granuloma is a focal aggregate of immune cells that forms in response to a persistent inflammation to encapsulate foreign material [4]. Formation of granulomas in GPA is a complex and not fully understood process involving both innate and adaptive immune responses. In genetically predisposed patients, exposure to a combination of various risk factors (among them aging, colonization with S aureus, nasal microbiome, and silicone exposure) leads to dysregulation of B and T cell balances. The active inflammatory process leads to the expression of proinflammatory cytokines and the formation of tertiary local lymphoid tissue (TLS) [5,6] in mucosal lesions [7]. Within these lesions, Voswinkel et al. reported sites of activation and maturation of PR3-positive B-cells, resembling memory B-cells [8]. Several studies had reported that peripheral blood obtained from patients with PR3-positive GPA contains mutated and autoreactive PR3-expressing B cells, which have, in yet unknown mechanism, escaped peripheral control checkpoints [9]. It has been suspected that the initial formation of lymphocytic cell aggregates may promote, via a positive feedback loop, the presentation of ANCA antigens to T cells, thereby leading to a Th1 shift [8] and the enhanced production of ANCA by B cells in situ [9]. Although the inflammatory process seems to be triggered by ANCAs, the pathophysiology involved in disease progression seems to differ between the formation of granulomas and vasculitis symptoms.

The exact mechanism of granuloma formation is unknown but researchers postulate the dominant role of abnormal T-cell signaling [10], by activating the macrophages through toll-like receptors (TLR) [5]. A recent study by Henderson et al. revealed that, indeed, PR3 initiates the formation of multinucleated giant cells from monocytes and macrophages and the formation of granuloma-like structures surrounded by T cells [11]. Further, the accumulation of monocytes and macrophages followed by subsequent expression of TNF and IL-1β [12] supports the influx of neutrophils [13,14] in a process described as ‘neutrophil swarming’. Moreover, a simultaneous shift towards Th17 cells from naive-T cells and the overexpression of IFN-g, TNF, IL-6, and Blc-2, by cytotoxic CD28 T-cells [15,16,17], further increases the proinflammatory burden. Furthermore, the anti-apoptotic effect of BLC-2 on Th1 cells also promotes the overexpression of proinflammatory cytokines. Current research has revealed an “amplification loop” for ANCA-mediated neutrophil activation via the C5a-complement protein [2,7]. Although an alternative complement pathway activation results in overstimulation by C3b, C5a, and C5b, which are also potent immune mediators [18], and the formation of membrane attack complexes, which result in the injury of targeted membrane structures of involved organs. Moreover, a positive feedback loop, via Cd5, enhances further activation of innate and adaptive immune responses, which finally, leads to the development of systemic vasculitis symptoms [18]. This positive feedback loop also leads to the formation of micro-abscesses and the generation of apoptotic cells, necrotic debris, and geographical necrosis, which characterize granulomas [14].

Understanding the differences in the pathophysiology between the formation of granulomas and vascular inflammation may be crucial to understanding the varied clinical course of GPA. Due to the varied and potentially life-threatening course of the disease, factors involved in poor prognoses have a significant role for clinicians. Risk factors for increased mortality include older age, high BVAS (Birmingham Vasculitis Activity Score), kidney failure, acute respiratory failure, gastrointestinal bleeding, and acute cardiovascular incidents [14,19,20,21,22,23]. Factors involved in poor prognoses, however, are not necessarily factors of a poor response to treatment. Signs originating from vascular inflammation, i.e., pulmonary infiltrates, nasal crusting, and renal vasculitis, are usually successfully treated by immunosuppressive treatments, while symptoms that originated from locoregional masses were described as highly resistant to treatment [3]. ENT involvement, including the temporal bone, is associated with a good prognosis, although has a high relapse rate [20,23,24]. In our article, we propose to add another risk factor of poor prognosis, which is skull base involvement. We present three cases of GPA, which meet the ACR 1990 and ACR/EULAR 2022 criteria for cANCA-positive GPA with typical histopathological changes obtained from ENT regions, with infiltration of the temporal bone and the base of the cranium. Despite adequate response to initial treatment, GPA locoregional involvement in all three patients remained resistant to immunomodulatory therapy and developed into a potentially life-threatening disease.

## 2. Case Reports

### 2.1. Case 1

A 68 years old, PR3-ANCA positive, male was initially diagnosed with non-organ threatening GPA based on the ACR 1900 criteria: typical sinus changes, nasal strands (Figure 1A), lung infiltrations in HR-CT, and polyneuropathy, cANCA 1455 CU (range: 0–19 CU).

The patient also complained of myalgia, nasal crusting, and pain with hearing loss in the left ear. At that time, no changes in the nasopharynx were observed (Figure 1B).

The patient was initially treated with methylprednisolone (MP), 40 mg/per day, and cyclophosphamide (CYC) in pulses of 500 mg every two weeks (a total dose of CYC 3 g), with a good clinical response, regression of pulmonary infiltrates, and the disappearance of myalgia and nasal crusting along with a reduction in cANCA (146.6 CU, range: 0–19 CU). After a reduction in MP dose, below 24 mg/day, the patient reported severe pain, which was localized near the left mastoid area. During hospitalization, the patient developed VI, VII, and VIII cranial nerve palsies with cerebellar symptoms and a severe headache. Additionally, jugular foramen symptoms were evolving and manifested in hoarseness and swallowing disorders (X and XI cranial nerve). CT and MRI scans revealed rapid progression of inflammatory mass, which was infiltrating the whole temporal bone, cerebellum, cerebellopontine angle, and nasopharynx, with concomitant cerebellar sinus thrombosis (Figure 1C–F).

The patient received MP pulses (3 × 500 mg, every 2 weeks—3 cycles sum of MP 4500 mg) with partial clinical improvement—resolution of pain, cerebellar symptoms, and VIII nerve palsy. However, no regression of the granulomatous mass in the skull base was noted. The clinical improvement was only temporal as the previously described symptoms returned in the following two weeks after the last MP pulses. We started treatment with rituximab (RTX) in a scheme 4 × 375 mg/m^2^ every week, continued CYC (up to a total dose of 4700 mg) with pulses of MP (4 g in 4 days), followed by maintenance therapy with prednisone 1.5 mg/kg/day, and mycophenolate mofetil (MMF) (3 g/per day). Additionally, the patient received local treatment of intratympanic 4 mg dexamethasone injections once per week for 6 weeks. After that treatment, the patient finally reached clinical remission and regression of inflammatory mass, as seen in CT scans of the temporal bone (Figure 1G).

### 2.2. Case 2

A 19-year-old male, diagnosed at the age of 14, with a juvenile PR3–ANCA positive GPA, was admitted to our department due to a relapse of the disease. The disease was diagnosed based on the ACR 1990 criteria: the presence of cANCA (1666 CU—above methods sensitivity, normal range: 0–19 CU), and life-threatening organ clinical symptoms, including pauci-immune glomerulonephritis with kidney failure (involving 80% of glomeruli in renal biopsy), involvement of the upper respiratory tract with the destruction of paranasal sinuses, and cranial with nerve palsies. The initial induction treatment included glucocorticoid steroids (GCs), immunoglobulins, plasmapheresis, and CYC, at an initial dose per os (2 mg/kg, total dose 5 g), although due to exacerbation of the disease during GCs, tampering changed to IV pulses (500 mg every two weeks, total dose including previous per os treatment—11 g). Due to a deteriorating kidney function, a progression of pulmonary inflammatory lesions observed in the CT scan, and a rise in the cANCA levels (cANCA 1200 CU), concomitant therapy with RTX (375 mg/m^2^ for 4 consecutive weeks) was introduced, and at the age of 16, the patient had reached a clinically stable disease condition. The maintenance therapy was GCs (prednisone, 5–30 mg/day) and methotrexate (MTX) 25 mg/week, yet due to a mild exacerbation, the MTX was changed to MMF 2 g/day. After 2 years of remission, at the age of 18, the patient reported increasing nasal crusting and mild otalgia with a progressive loss of hearing. Using MR, a massive inflammation was found infiltrating the temporal bone (Figure 2A).

The cANCAs level was 90.2 CU. Due to the GPA relapse, induction therapy with CYC (summary dose of cycle 6 g, total life dose of 17 g) with MP pulses was reintroduced. Additionally, the patient underwent a surgical tympanostomy but showed only minor improvement. Owing to the radiological progression of the inflammatory mass in the temporal bone and the progression of hearing impairment, followed by a rise in ANCA levels (125.6 CU), the patient received RTX in 4 pulses of 325 mg/m^2^, which prompted substantial clinical improvement. The maintenance therapy consisted of GCs (prednisone, 1 mg/kg/day), MTX (25 mg/week), and RTX (1000 mg every 6 months) and was supported by 4 tympanostomies with drainage followed by intratympanic dexamethasone injections (4 mg each). The treatment resulted in the stabilization of the clinical state but osteoneogenesis in the temporal bone, the mastoid process, and sinuses showed no significant changes in the CT scan (Figure 2B).

### 2.3. Case 3

A 34-year-old female with GPA was admitted to our department due to progressive loss of hearing, severe otalgia, nasal crusting, and mild cerebellar symptoms. A diagnosis of GPA was made when the patient was 29 years old based on the ACR 1900 criteria: inflammatory tumor and cavitations in the right lung, tracheostenosis, paranasal sinuses destruction, rapidly progressing “saddle nose”, fever, polyneuropathy, and the presence of cANCAs (966 CU, range: 0–19 CU). CT and MRI scans also revealed an inflammatory mass surrounding the pituitary gland as well as bilateral mastoiditis with infiltrations of the inner ear. The patient was initially treated with CYC pulses (total dose of 5.6 g) and MP pulses (3 × 500 mg) followed by GCs (prednisone, 1 mg/kg/d). After induction therapy, the patient achieved a regression in the pulmonary changes, with a recession in the nasal crusting, improvement of neurological symptoms, and lowering of cANCa levels (196.2 CU, range: 0–19 CU). However, despite maintenance treatment with MTX (25 mg/week), which was later changed to azathioprine (150 mg/day) owing to poor clinical toleration to MTX, while continuing a high dose of GCs (prednisone, 30–60 mg per day for 6 months), multiple surgical ear interventions and intratracheal GC local injections (MP 40 mg, 10 times in 12 months), progression of the inflammatory mass in both temporal bones was observed in CT scans (Figure 3A), followed by the loss of hearing in the right ear with recurrent otorrhea, severe otalgia, and progressing chronic cerebellar symptoms with 6th cranial nerve palsies.

Due to the progression of inflammation in the base of the skull, treatment with MP pulses (3 × 500 mg) followed by prednisone 1 mg/kg/day, MMF (2 g/day), and RTX (4 × 375 mg/m^2^/week) was introduced, with concomitant intratympanic injection of 4 mg of dexamethasone (1/week for 3 months). Local treatment with tympanostomy tube insertion and local treatment with steroids improved the hearing loss and was followed by withdrawal of the temporal bone changes in the CT scan (Figure 3B).

## 3. Discussion

GPA with skull base involvement seems to be a distinct entity of the locoregional form of GPA, with potentially severe disease course and resistance to the treatment. The frequency of cranial base involvement is estimated to be 3–10% of GPA [25,26,27]. Due to insidious progress, it can be easily misinterpreted as regional ear involvement. In the presented cases, skull base involvement in GPA combines organ-threatening ENT disease with additional CNS symptoms. All our patients reported hearing loss and different cranial nerve palsies, with additional symptoms similar to cerebellopontine angle syndrome, while in one case (case 1), in jugular foramen syndrome. Interestingly, despite the locally aggressive course of the disease, ANCA levels in our patients were significantly lower than at the time of the diagnosis and initial treatment. This is most likely because of the different involvement of ANCAs in the pathogenesis of the formation of extravascular necrotizing granulomas from vasculitis symptoms [16]. The GPA with locoregionally advanced disease is distinguished not only by the severity of the clinical symptoms of the patients but also by the resistance to pharmacological treatment [28]. In all our cases systemic disease activity decreased after induction therapy with immunomodulatory drugs (there was a resolution of pulmonary, renal, nasal, cutaneous, or peripheral nervous symptoms). However, the patients’ locoregional symptoms persisted, and local inflammatory infiltration progressed despite treatment.

Choosing the right regimen of immunomodulating therapy for GPA with cranial base involvement remains a challenge. Local disease progression can be less responsive to MTX and high doses of GCs than other GPA manifestations [14]. Among our patients, in two cases, CYC therapy in combination with GCs was ineffective, as the disease was exacerbated during the reduction in GCs, despite concomitant immunosuppressive treatment with MTX and MMF. Similar results were observed in previous reports in patients initially treated with CYC and high doses of MP [3,28,29]. According to the literature, even in cases with an initial good response to CYC with MP [30,31], almost 50% of patients relapsed within a year [31]. Following the recent data, treatment with RTX shows mixed results in patients with granulomatous manifestations in the head and neck region. Studies and case studies have shown both poor [32,33,34,35] and good responses after RTX in monotherapy [36,37].

Similar to the results presented by Cortazar et al. [37], McAdoo et al. [38], Peters et al. [32], and Kim et al. [39], in our cases 1 and 2, treatment with CYC with MP pulses and subsequent RTX was successful in reducing clinical activity and stopping the progression of bone infiltrations in the MRI/CT scans. These results suggest complementary effects of combined CYC and RTX in GPA [14]. As CyC and RTX show different mechanisms of action, it could be hypothesized that synergism of T cell CyC-induced apoptosis and concomitant CD 20(+) B cells depletion, mediated by RTX, could affect both pathways responsible for the development of GPA. However, it has to be highlighted that despite the effectiveness of combined treatment, in none of our cases did we observe full resolution of inflammatory lesions. Another issue is the possible toxicity of combined therapy with RTX and CYC. In our cases, we did not observe any serious adverse events, although there are no other studies to confirm it, as in previously reported cases, authors used CYC in lower doses.

In the described cases, apart from systemic treatment, patients also received local treatment. Intratympanic steroid injection is typically administered in the treatment of Meniere’s disease or idiopathic sudden sensorineural hearing loss [40,41] and has not been described as a standard procedure in GPA. In our case, due to massive inflammation in the temporal bone with progression in the base of the skull, we decided on a “rescue treatment” with dexamethasone intratympanic injections, theorizing that local injections with glucocorticoid steroids may be effective in limiting inflammation in temporal bone infiltration. GCs were applied intratympanically using needles by an otorhinolaryngologist. We have chosen dexamethasone due to its long effect, high potency, and high concentration in a minimal volume of fluid. In one case (case 2) application of dexamethasone was administered after tympanothomies. We have found only one similar case report, by Kim et al. [39], who described a patient with severe inner ear effusion presenting as sudden deafness in the course of GPA. The patient had a severe systemic form of GPA that required treatment with RTX (4 × 500 mg every 2 weeks) with concomitant CYC (3 g iv) and pulses of MP (3 × 500 mg/day). Due to severe otitis, the authors decided on a follow-up treatment with intratympanic GCs injections, which resulted in an improvement in hearing. Despite similarities, in our cases, treatment-resistant disease progression was associated with the bulk of the tumor localized in the temporal bone, which is a distinct clinical manifestation of otitis.

A recent emerging approach in GPA treatment is adjunctive therapy with direct neutrophil activation block along targeted therapies of T cells and B cells. Avacopan is a novel small-molecule blocker of neutrophil chemoattraction and activation via C5a receptor blockade [30]. In the 2021 phase 3 clinical study “ADVOCATE”, avacopan demonstrated its noninferiority to GCs in the treatment of vasculitis-delivered AAV symptoms. The aforementioned study revealed that avacopan efficiently reduced steroid uptake along with an acceptable safety profile, in combined treatment with standard treatment, and for both CyC and RTX [31]. Despite promising outcomes, none of the patients enrolled in CLEAR, CLASSIC, and ADVOCATE studies on avacopan were diagnosed with inflammatory pseudotumor or severe locoregionally aggressive CNS involvement. The clinical utility of avacopan in patients with cranial base involvement has yet to be tested. Therefore, we hope that further research will allow the development of tailored treatments for GPA with skull base involvement.

Moreover, we have compared our cases to the available literature. We have searched for case reports and case series, yet we have only found several that are applicable [42,43,44,45,46,47], as we had to exclude cases without histopathological confirmation, infection, suspicion of malignancy, and lacking in data or describing other limited disease manifestations in the head and neck regions. All the presented cases are compared in Appendix A. Interestingly our cases were diagnosed much later, meaning loco-aggressive symptoms had developed over time, while in the available cases, locoregional disease progressions were the primary cause for diagnostics. There are several differences, in the presented cases, authors reported the resolution of the inflammatory masses and symptoms, while in our cases, we have mostly managed to stabilize disease progression and achieved the partial dissolution of symptoms in cases 2 and 3 and a full recovery in case 1. Furthermore, a reduction in cANCA levels was observed despite life-threatening disease progression, thereby potentially strengthening the hypothesis that there are differences in the pathophysiology between the development of granulomas and vasculitis symptoms. Yet, we cannot compare this observation to others in the literature or case series because the authors did not report a follow-up.

## 4. Conclusions

GPA with skull base involvement presents a distinct disease phenotype. According to our experience and available literature, skull base involvement in GPA is associated with poor clinical outcomes and low response rates to classical monotherapy with CYC and GCs. The insidious nature of the skull base involvement in the course of GPA together with its resistance to induction therapy may result in potentially life-threatening conditions. In all presented cases, treatment with RTX led to clinical stabilization but not a full resolution of inflammatory lesions, which supports a need for further studies on combined treatment in GPA. We propose sequential therapy based on CYC and RTX with additional GCs (both systemic and local) for the most treatment-resistant cases.

## Figures and Tables

**Figure 1 brainsci-13-01140-f001:**
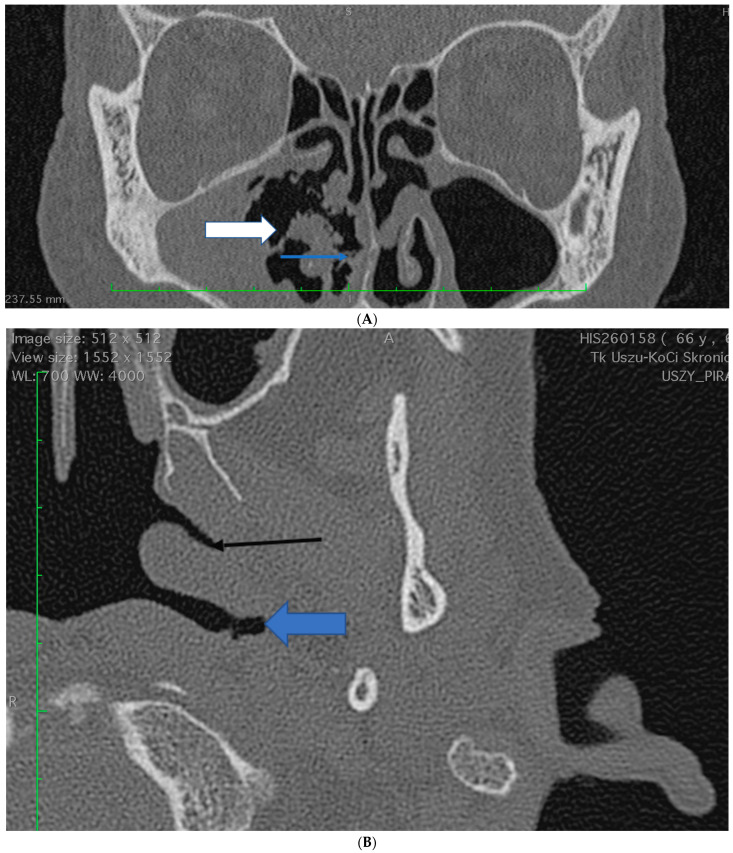
(**A**) Patient 1 (the initial stage of the disease). CT scan: osteitis/osteoneogenesis of the right maxillary sinus (the difference in the width of superior and inferior maxillary sinus wall on both sides), destruction of the lateral nasal (white arrow and blue arrow). (**B**) Patent 1 (the initial stage of the disease). CT scan: Eustachian tube (black arrow) and marked Rosenmuller fossa (wide, blue arrow). (**C**) Patient 1. CT scan: Progression of changes: Inflammatory mass in the left pterygopalatine fossa (wide blue arrow)—obstruction of anatomical landmarks—Eustachian tube, Rosenmuller fossa, and right maxillary sinus—partial improvement with persistent tissue destruction of the lateral nasal wall and osteitis of other walls. (**D**) Patient 1. MRI scan: Fluid in the mastoid process (wide, blue arrow), inflammatory process on the skull base reaching the jugular foramen posteriorly and laterally (white frame). (**E**) Patient 1. MRI scan: Mass on the skull base, in the nasopharynx (blue arrows 2.67 × 2.48 cm) spreading below the petrous and lacerum portion of the carotid artery). (**F**) Patient 1. CT scan: Resolution of the inflammatory tumor in the left pterygopalatine fossa (blue arrow) compared to Figure 1C. (**G**) Patient 1. MRI scan after the treatment: the resolution of the inflammatory process on the skull base (blue arrow) compared to Figure 1D.

**Figure 2 brainsci-13-01140-f002:**
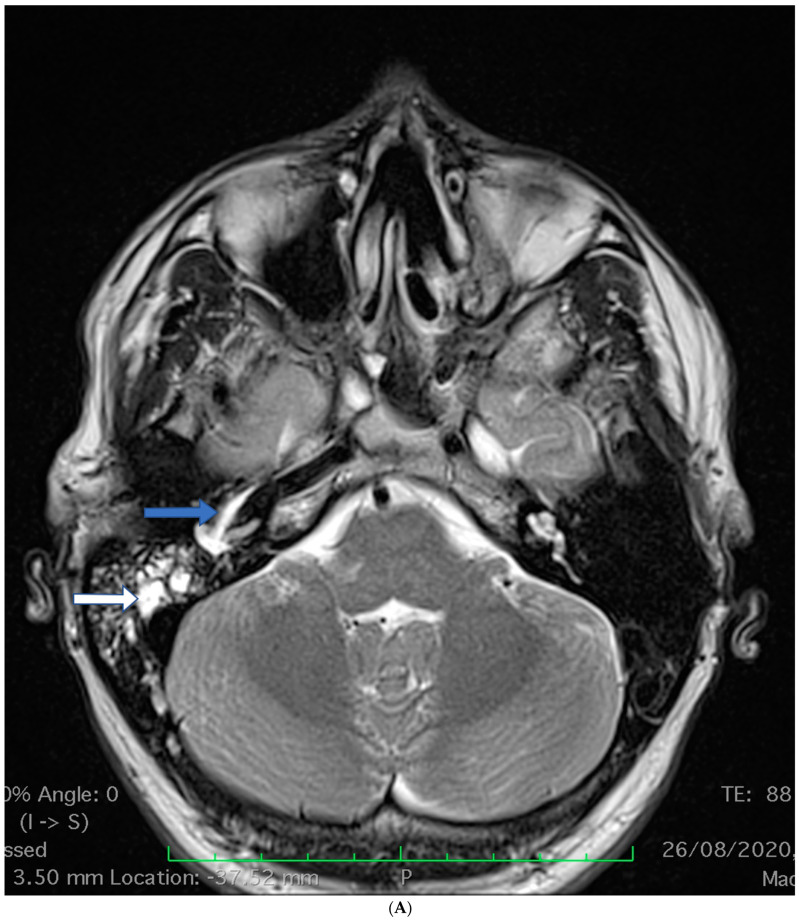
(**A**) Patient 2. MRI—fluid in the right temporal bone—the mastoid process (white arrow), tympanic cavity, and filling the bony part of the Eustachian tube close to the carotid artery (blue arrow). (**B**) Patient 2. CT—right temporal bone osteomyelitis (white frame).

**Figure 3 brainsci-13-01140-f003:**
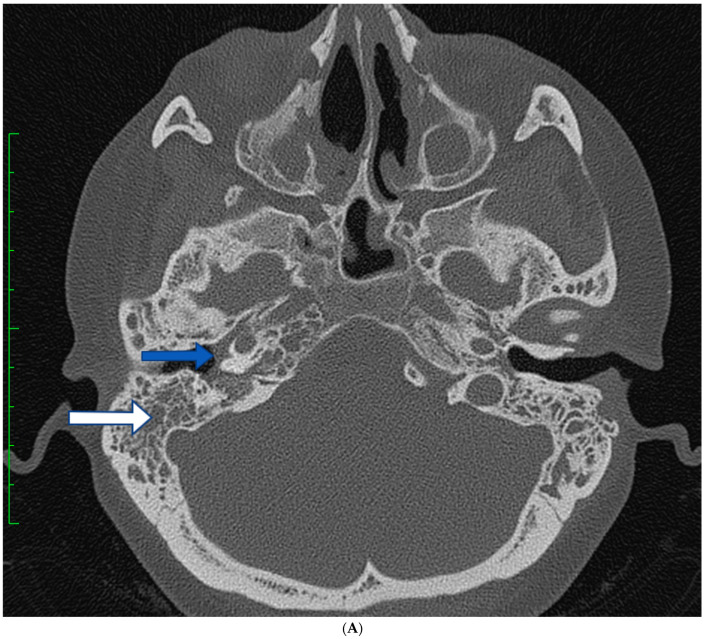
(**A**) Patient 3. CT scan of the temporal bones—complete opacification of the mastoid process and tympanic cavity (white and blue arrows, respectively). The blue arrow also points to the pull in the tympanic membrane. (**B**) Patient 3. CT of the temporal bones after the treatment—blue arrow shows the ear drain used to locally administer dexamethasone and the white arrow shows the mastoid process without opacification.

## Data Availability

The data underlying this article are available in the article and its online Appendix A.

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
