# Peer review of "Combination Treatment of Locoregionally Aggressive Granulomatosis with Polyangiitis and Cranial Base Infiltration"

_brainsci, 2023, doi:10.3390/brainsci13081140_

Round 1

Reviewer 1 Report

This is an interesting paper offering the management of three patients with a localized ENT GPA form.

However, some points need to be more explained and corrected:

The title was not appropriate. It is not a case sequence therapy with CYC and RTX; there were a lot of different treatments given, not all were treated by local DM.

Please correct the Title… it can only be something like “Management of a progressive (localized) GPA – report of three cases..” or similar...

Abstract:

The abstract should start with a definition of GPA but not with the objective, so please reorganize this part.

Please kindly note that this is only a systematic review if the authors provide the whole procedure of a process and methodology to search different bases, articles, numbers of found, excluded articles, and so on. However, the authors should provide a table with all previously found cases on this subject if it is a literature review.

The therapy modalities differed for all cases, so the phrase about the therapy within the abstract should be more precisely written.

This part should be rephrased as “less likely to respond to pharmacotherapy” because the author also provides data regarding the use of pharmacotherapy but not other treatment options…

Introduction:

3rd line of the 1st paragraph: a term ANCA should be introduced correctly

12th  line of the 1st paragraph: How many cases do you present?

Please don't start a sentence with abbreviations and numbers; correct this throughout the text.

Case 1

More data are needed: titer of ANCA on IIF or concentration if measured using the ELISA test. The authors should mention when the diagnosis was made (disease duration) to follow the disease's whole course.

It is pretty unusual for a patient to present with an ILD within this phenotype of AAV (GPA). So the authors have to provide more data on this (pattern of ILD, changes seen such as localization, morphological pattern (honeycombing, bronchiectasis) or even to put a figure of an HRCT, lung function tests (FVC, DLCO)? Did the authors take into consideration antifibrotics? How was the EGPA excluded? What were the results of the CBC and inflammatory parameters? Which criteria were applied for the diagnosis?

7th  line of the 1st paragraph: Considering that the authors described an ILD at the beginning, how it comes to regression on pulmonary infiltrates (this is the other form of lung disease?)

The authors must be more precise with a therapy: how many cycles in a total of cyclo?

Was there any bridging immunomodulatory therapy after the completion of CYC cycles? How long from the last cycle was MRI done, revealing a local progression?

So, how many pulses of CYC did the patient receive? The authors must provide more data on when the RTX was started; what does "4g in 4 days MP" mean? (did the patient have this dose of MP in 4 subsequent days?), and most importantly, what was the approach, and how was the control of the DM application possible? Who performs this application of DM? Is it a needle installation of MP? What was the dose of DM? Is it a single dose of MP? As a reader, I would be especially interested in this part of the case presentation since it is not a routinely performed procedure.

Case 2:

More data are needed: titer of ANCA on IIF or concentration if measured using the ELISA test.

Could the author provide the exact timeline with the treatment applied: CYC (cumulative dose), RTX, MTX, MMF during the period of remission lasting two years? What was the rationale for changing the treatment if the remission was achieved? Is it 6g of CYC in total, including CYC cycles at the beginning of the disease? According to the clinical course and the treatment applied, the patient was not treated with the local application of DM.

Case 3:

The same for the third case: More data on the titer of ANCA on IIF, or concentration if measured using the ELISA test, should be provided

Could the authors explain what  "5.6g/one cycle" means? The treatment has to be explained more precisely over time. What was the rationale for changing MTX to AZA, and what was the dose of AZA per kg BW?

Please, provide the exact timeline of the treatment: “Treatment with MP pulses (3x500mg) followed by 1mg per kg prednisone and MMF (2g/day) with RTX was introduced (4x375mg/m2/week) and intratympanic injection of 4mg of dexamethasone (1/week for three months).

I suggest providing the timeline with the treatment applied for all cases on the same graph or putting all the data in the table for all three cases in a way to be comparable.

Figures

All figures for each patient should be mixed up in a panel, including before and after the treatment. All figures MUST be provided with arrows aiming to give the exact location mentioned in the legends for each figure.

English should be carefully checked and corrected.,

Please use the same abbreviation throughout the text, and be consistent with them. What is the difference between GCS vs. Gcs (6th line of the first paragraph, Case 2)

METHODOLOGY FOR A LITERATURE REVIEW WAS COMPLETELY MISSED! If it is a literature review, please provide a methodology and previously published cases on this subject!

Discussion

Line 12 in the 1st Paragraph:  “Also, during the disease in our patients, ANCA antibodies

levels remained low.” Could you explain this, low – below the reference level, or low positive, and the rationale for applying RTX for “low” ANCA (not needed to include antibody after ANCA; it is included within the abbreviation.

The discussion should be comprehensive and have a thorough literature review. I suggest you have all previously published similar cases in a separate table. More of it is needed to support the decision to install DM locally in Case 1 and Case 3.

The conclusion should be rewritten: all cases are different, and it is impossible to maintain them under the same umbrella (GCS pulses with CYC, followed by the instant transition to RTX followed by local treatment with GCS, was the most effective therapeutic option). Moreover, the second was not provided with the local DM.

Need to be improved, provide a professional or a native English speaker in purpose to check this issue

Author Response

Dear Reviewer,

thank You for your Review,we have revised manuscript according to Your suggestions

The abstract should start with a definition of GPA but not with the objective, so please reorganize this part.

The Abstract has been reorganized according to reviewers suggestions

Please kindly note that this is only a systematic review if the authors provide the whole procedure of a process and methodology to search different bases, articles, numbers of found, excluded articles, and so on. 

We agree with reviewers note that our manuscript is not a systematic review thus we have modified title and manuscript

However, the authors should provide a table with all previously found cases on this subject if it is a literature review.

Table was added of manuscript

The therapy modalities differed for all cases, so the phrase about the therapy within the abstract should be more precisely written.

As follows, abstract has been reorganized and rewritten according to reviewers suggestions

This part should be rephrased as “less likely to respond to pharmacotherapy” because the author also provides data regarding the use of pharmacotherapy but not other treatment options…

It has been rephrased as reviewer suggested

Introduction:

3rd line of the 1st paragraph: a term ANCA should be introduced correctly

this mistake has been corrected

12th  line of the 1st paragraph: How many cases do you present?

We present three cases, I apologize for mistake, it had occured due to several text reworks

Please don't start a sentence with abbreviations and numbers; correct this throughout the text.

It has been corrected according to reviewers suggestion

Case 1

More data are needed: titer of ANCA on IIF or concentration if measured using the ELISA test. The authors should mention when the diagnosis was made (disease duration) to follow the disease's whole course.

This issue has been answered by adding cANCA level measured using Elisa with normal ranges. Disease duration has been added in table 1 for comparability. In all cases treatment and disease duration time periods had been added  

It is pretty unusual for a patient to present with an ILD within this phenotype of AAV (GPA). So the authors have to provide more data on this (pattern of ILD, changes seen such as localization, morphological pattern (honeycombing, bronchiectasis) or even to put a figure of an HRCT, lung function tests (FVC, DLCO)? Did the authors take into consideration antifibrotics? How was the EGPA excluded? What were the results of the CBC and inflammatory parameters? Which criteria were applied for the diagnosis?

This is of course a mistake made by our group, patients were not diagnosed with ILD and it has been corrected. Diagnosis of GPA was stated based on ACR 1990 Criteria, but all patients retrospectively match ACR/EULAR 2022 Criteria. Also this information had been added to the main body. Patients did not meet the criteria of EGPA and histopatological findings from ENT regions were typical for GPA

7th  line of the 1st paragraph: Considering that the authors described an ILD at the beginning, how it comes to regression on pulmonary infiltrates (this is the other form of lung disease?)

As follows, term ILD was used mistakenly and we apologize for it.

The authors must be more precise with a therapy: how many cycles in a total of cyclo?

This issue has been answered by adding exact doses of cyc and total life dose.

Was there any bridging immunomodulatory therapy after the completion of CYC cycles?

In cases 1 and 2 there was no bridging therapy due to fast progression of clinical symptoms. In case 3 we present maintenance therapy with MTX and AZA, although it was not effective.

How long from the last cycle was MRI done, revealing a local progression?

In case 1 MRI was performed during CyC cycle and RTX was added concomitantly with CyC, what has been added to case for better clarity.

So, how many pulses of CYC did the patient receive? The authors must provide more data on when the RTX was started; what does "4g in 4 days MP" mean? (did the patient have this dose of MP in 4 subsequent days?),

We have addressed thi issue by rephrasing for better clarity cases

In case 1 it was rephrased: 

The patient was initially treated with methylprednisolone (MP) 40 mg/per day and cyclophosphamide (CYC) in pulses 500mg every two weeks (a total dose of CYC 3g) with a good clinical response, regression of pulmonary infiltrates and the disappearance of myalgia and nasal crusting with lowering of cANCA (146,6 CU range 0-19 CU).”

In case 2:

“and CYC - initially per os (2mg/kg, total dose 5g), but due to exacerbation of the disease during GCs tampering changed to IV pulses (500mg every two weeks, total dose including previous per os treatment – 11g)” and “Due to the GPA- relapse, induction therapy with CYC (summary dose of cycle 6g, total life dose of 17g) with pulses of MP was reintroduced.”

and most importantly, what was the approach, and how was the control of the DM application possible? Who performs this application of DM? Is it a needle installation of MP? What was the dose of DM? Is it a single dose of MP? As a reader, I would be especially interested in this part of the case presentation since it is not a routinely performed procedure.

In accordance with the reviewer we have added following data:  clarified that steroids were dexomethasone  4mg with number of dasages. GCs were applied intratympanic using needle  by an otorhinolaryngologist in cases 1 and 3 and in case 3 through drainage tube.

Case 2:

More data are needed: titer of ANCA on IIF or concentration if measured using the ELISA test.

This issue has been answered by adding cANCA level measured using Elisa with normal ranges. Disease duration has been added in table 1 for comparability. In all cases treatment and disease duration time periods had been added  

Could the author provide the exact timeline with the treatment applied: CYC (cumulative dose), RTX, MTX, MMF during the period of remission lasting two years? What was the rationale for changing the treatment if the remission was achieved? Is it 6g of CYC in total, including CYC cycles at the beginning of the disease? 

According to reviewer notes time periods were added in text for better clarity

According to the clinical course and the treatment applied, the patient was not treated with the local application of DM.

As reviewer noted, during editions of the text we have omitted information that the patient received dexamethasone intratympanic, as we have stated in introduction and abstract. We apologize for the lack of clarity on this matter

 Case 3:

The same for the third case: More data on the titer of ANCA on IIF, or concentration if measured using the ELISA test, should be provided

This issue has been answered by adding cANCA level measured using Elisa with normal ranges

Could the authors explain what  "5.6g/one cycle" means? The treatment has to be explained more precisely over time. What was the rationale for changing MTX to AZA, and what was the dose of AZA per kg BW?Please, provide the exact timeline of the treatment: “Treatment with MP pulses (3x500mg) followed by 1mg per kg prednisone and MMF (2g/day) with RTX was introduced (4x375mg/m2/week) and intratympanic injection of 4mg of dexamethasone (1/week for three months).

this issue has been answered by rephrasing case 3:

“The patient was initially treated with CYC pulses (total dose of 5.6g) with MP pulses (3x500mg) followed by GCs (prednisone, 1mg/kg/d). After induction therapy the patient achieved regression of pulmonary changes, receding of nasal crusting, improvement of neurological symptoms, and lowering in cANCa levels (196,2 CU range 0-19 CU). However, despite maintenance treatment with MTX (25 mg/week), later changed to azathioprine (150mg/day) due to poor clinical toleration of MTX, while continuing a high dose of GCs (prednisone, 30-60 mg per day for 6 months), “

Figures

All figures for each patient should be mixed up in a panel, including before and after the treatment. All figures MUST be provided with arrows aiming to give the exact location mentioned in the legends for each figure.

this is an obvious mistake, witch occurred during data transfer and we have corrected it

English should be carefully checked and corrected.,

Please use the same abbreviation throughout the text, and be consistent with them. What is the difference between GCS vs. Gcs (6th line of the first paragraph, Case 2)

This issue has been corrected and unified terminology

METHODOLOGY FOR A LITERATURE REVIEW WAS COMPLETELY MISSED! If it is a literature review, please provide a methodology and previously published cases on this subject!

We agree with the reviewer that we have overstated that our manuscript is a review or systematic review, it has been corrected in the whole text.

Discussion

Line 12 in the 1st Paragraph:  “Also, during the disease in our patients, ANCA antibodies

levels remained low.” Could you explain this, low – below the reference level, or low positive, and the rationale for applying RTX for “low” ANCA (not needed to include antibody after ANCA; it is included within the abbreviation.

We agree with reviewer, that our statement was unclear and we have rephrased it : Interestingly, despite the locally aggressive course of the disease, ANCA levels in our patients were significantly lower than at the time of diagnosis and initial treatmet

The discussion should be comprehensive and have a thorough literature review. I suggest you have all previously published similar cases in a separate table. More of it is needed to support the decision to install DM locally in Case 1 and Case 3.

According to reviewer following lines were added:” In the described cases, apart from systemic treatment, patients also received local treatment. Intratympanic steroid injection is typically administered in the treatment of Meniere's disease or idiopathic sudden sensorineural hearing loss [44] and has not been described as a standard procedure in GPA. In our case, due to massive inflammation in the temporal bone with progression in the base of the skull, we have decided on a “rescue treatment” with dexamethasone intratympanic injections, theorizing that local injections with glucocorticosteroids may be effective in limiting inflammation in temporal bone infiltration. GCs were applied intratympanic using needles by an otorhinolaryngologist. We have chosen dexamethasone due to its long effect, high potency, and high concentration in a minimal volume of fluid”

The conclusion should be rewritten: all cases are different, and it is impossible to maintain them under the same umbrella (GCS pulses with CYC, followed by the instant transition to RTX followed by local treatment with GCS, was the most effective therapeutic option). Moreover, the second was not provided with the local DM.

We agree with reviewer that the cases are not comparable and there were significant differences in treatment modalities, thus the conclusion had been rewritten into more precise 

Reviewer 2 Report

The authors described three patients with granulomatosis with polyangiitis (GPA) with skull base involvement and did not respond to the initial combination therapy with cyclophosphamide and glucocorticosteroids but responded to subsequent rituximab therapy. The review of the literature suggested worse clinical and radiological response to standard therapy in GPA patients with skull base involvement.

This is an interesting article providing important insights into current knowledge on the management of GPA. Multidisciplinary approach is needed for GPA because it is a systemic disease. Hence, this article will attract broad range of readers. The manuscript is well written and I enjoyed reading it.

Minor issues and suggestions to strengthen this manuscript are raised as follows: 

1. I would recommend including issues regarding emerging therapies for ANCA-associated vasculitis including GPA. For example, an efficacy of avacopan has recently been demonstrated by a phase III study involving patients with GPA (N Engl J Med 2021; 384: 599-609).

2. Recent articles focused on the importance of ANCA and granulomatous inflammation in the pathophysiology of GPA (Int J Mol Sci 2021; 22: 6474; Neurol Ther 2022; 11: 21-38). This issue should be incorporated in the introduction section by citing these articles to facilitate comprehension of pathophysiology of GPA for readers.

3. “We present five cases of GPA” in the introduction section would be “We present three cases of GPA”.

Author Response

Dear Reviewer,

thank You for your Review,we have revised manuscript according to Your suggestions

  1. I would recommend including issues regarding emerging therapies for ANCA-associated vasculitis including GPA. For example, an efficacy of avacopan has recently been demonstrated by a phase III study involving patients with GPA (N Engl J Med 2021; 384: 599-609).

According to reviewer suggestions we have included aforementioned study and its follow-ups on Acacopan as follows:”A recently emerging approach in GPA treatment is adjunctive therapy with direct neutrophil activation block along with T-Cells and B-cells targeted therapies. Avacopan is a novel small-molecule blocker of neutrophil chemoattraction and activation via C5a receptor blockade [34]. In the 2021 phase 3 clinical study “ADVOCATE”, avacopan has shown its noninferiority to GCs in the treatment of vasculitis-delivered symptoms of AAV. The aforementioned study revealed that avacopan efficiently reduced steroid uptake along with an acceptable safety profile in combined treatment with standard treatment with both CyC and RTX “.

  1. Recent articles focused on the importance of ANCA and granulomatous inflammation in the pathophysiology of GPA (Int J Mol Sci 2021; 22: 6474; Neurol Ther 2022; 11: 21-38). This issue should be incorporated in the introduction section by citing these articles to facilitate comprehension of pathophysiology of GPA for readers.

Following reviewer suggestions following lines were added, citing aforementioned research:

Neurological manifestations of GPA are less common, with peripheral nervous system involvement occurring in just 23% of GPA in comparison to 17-75% of all ANCA-associated vasculitis (AAV) [2]. Central nervous system (CNS) involvement occurs even less commonly, with only 3-6% of patients having granulomas within skull bones “ 

Thank You for highlighting those important articles, such as one from Int J Mol Sci 2021; 22: 6474. We have found this article extremely interesting and it has been cited in our manuscript along with others by aforementioned authors.

  1. “We present five cases of GPA” in the introduction section would be “We present three cases of GPA”.

this mistake has been corrected

Reviewer 3 Report

General comment : this brief report is actually a small case reports series, and contrarily to the statement in the abstract no literature review. The three reported cases are interesting in a rare disease and are worth being reported. There are several flaws that should be explained and/or corrected.

1.       The title is misleading: one could read this as a real sequential treatment in all three patients (cyclophosphamide followed by rituximab + local glucocorticoids[GC]). Actually all patients received various combination therapies not always in accordance with the EULAR recommendations (ref 1).Patient 1: systemic GC + cyclophosphamide, followed by rituximab and than maintenance with mycophenolate mofetil and local dexamethasone / according to the guidelines this could have been methotrexate or mycophenolate mofetil + GC. Patient 2: had a life-threatening disease and received everything upfront, including rituximab and again rituximab after relapse / according to guidelines he could have received cyclophosphamide or rituximab and GC + plasmapheresis. Patient 3: received cyclophosphamide + GC as starting therapy, which is classical, than GC + methotrexate and azathioprine and later GC pulses , mycophenolate mofetil and rituximab + local GC. I can understand that patient care often requires deviations from guidelines but the title does not represent what has actually been done.

2.       The images are generally of poor quality and not well ordered. Figure 1: I miss F, G, H. A seems to have been ‘elongated’, B is of poor quality and hard to understand, C lacks soft tissue contrast, D is poor. Figure 3 F is not described in the legend. The arrows indicated in the legend (which would considerably help clinicians with little expertise in neuroradiology) are missing on the images. Figure 2: A is mentioned in the text as MR and CT scan, it ia actually a poor contrasted MR. Conversley, 2B is mentioned in the text as and MR while it is a CT. Figure 3A refers to both temporal bones, when only the left one is shown. Figure 3B refers to CT and MRI when it is an MRI. Further the image is shrunk and of poor quality. In addition, all patient names should be removed from the images.

3.       On page 2, line 6, the authors mention five cases…

4.       On page 2 para 2, line one ‘organ-non-threatening GPA’ should read ‘non-organ threatening GPA’

5.       Reference 17 should read Jenette CJ

moderate improvement is needed

Author Response

Dear Reviewer,

thank You for your Review,we have revised manuscript according to Your suggestions

General comment : this brief report is actually a small case reports series, and contrarily to the statement in the abstract no literature review. 

We agree with the reviewer that we have overstated that our manuscript is a review or systematic review, it has been corrected in the whole text.

  1.       The title is misleading: one could read this as a real sequential treatment in all three patients (cyclophosphamide followed by rituximab + local glucocorticoids[GC]). Actually all patients received various combination therapies not always in accordance with the EULAR recommendations (ref 1).Patient 1: systemic GC + cyclophosphamide, followed by rituximab and than maintenance with mycophenolate mofetil and local dexamethasone / according to the guidelines this could have been methotrexate or mycophenolate mofetil + GC. Patient 2: had a life-threatening disease and received everything upfront, including rituximab and again rituximab after relapse / according to guidelines he could have received cyclophosphamide or rituximab and GC + plasmapheresis. Patient 3: received cyclophosphamide + GC as starting therapy, which is classical, than GC + methotrexate and azathioprine and later GC pulses , mycophenolate mofetil and rituximab + local GC. I can understand that patient care often requires deviations from guidelines but the title does not represent what has actually been done.

We agree with reviewer that our presentation of cases was not entirely clera, therefore we have added timelines, table for comparison reasons and preseted in chronological order applied treatment. also we have added clinical sympthoms to justify our treatment. We have further addressed this issue by rephrasing for better clarity cases

In case 1 it was rephrased: 

The patient was initially treated with methylprednisolone (MP) 40 mg/per day and cyclophosphamide (CYC) in pulses 500mg every two weeks (a total dose of CYC 3g) with a good clinical response, regression of pulmonary infiltrates and the disappearance of myalgia and nasal crusting with lowering of cANCA (146,6 CU range 0-19 CU).”

In case 2:

“and CYC - initially per os (2mg/kg, total dose 5g), but due to exacerbation of the disease during GCs tampering changed to IV pulses (500mg every two weeks, total dose including previous per os treatment – 11g)” and “Due to the GPA- relapse, induction therapy with CYC (summary dose of cycle 6g, total life dose of 17g) with pulses of MP was reintroduced.”

  1.       The images are generally of poor quality and not well ordered. Figure 1: I miss F, G, H. A seems to have been ‘elongated’, B is of poor quality and hard to understand, C lacks soft tissue contrast, D is poor. Figure 3 F is not described in the legend. The arrows indicated in the legend (which would considerably help clinicians with little expertise in neuroradiology) are missing on the images. Figure 2: A is mentioned in the text as MR and CT scan, it ia actually a poor contrasted MR. Conversley, 2B is mentioned in the text as and MR while it is a CT. Figure 3A refers to both temporal bones, when only the left one is shown. Figure 3B refers to CT and MRI when it is an MRI. Further the image is shrunk and of poor quality. In addition, all patient names should be removed from the images.

this is an obvious mistake made by our group. We appologize for it and we have corrected files according to reviewers notes.

  1.       On page 2, line 6, the authors mention five cases…

It has been corrected according to reviewers suggestion

  1.       On page 2 para 2, line one ‘organ-non-threatening GPA’ should read ‘non-organ threatening GPA’

It has been corrected according to reviewers suggestion

  1.       Reference 17 should read Jenette CJ

We apologize for this mistake, it has been corrected

Round 2

Reviewer 1 Report

Dear Authors,

The Revised form of the manuscript has been sufficiently improved. However, the figures are not in the document but have been enclosed in the supplementary files.

Could you please put the figures in the appropriate that they are supposed to be found within the manuscript and mixed up into the panel referred to each patient?

Zipp filles with all figures are not readable.. also adding it as a supplementary file means your intention is not to have them in the body of the manuscript

Reconsider after the Major revision

Author Response

Dear Reviewer,

thank You for Your suggestions.

However, the figures are not in the document but have been enclosed in the supplementary files.

Could you please put the figures in the appropriate that they are supposed to be found within the manuscript and mixed up into the panel referred to each patient?

Zipp filles with all figures are not readable.. also adding it as a supplementary file means your intention is not to have them in the body of the manuscript

Files has been added to the manuscript and in ZIP versions to keep quality of Figures

Reviewer 3 Report

Thanks to the authors for addressing the comments. The title is now more appropriate and the presentation of therapeutic sequence as a summarized table is useful. Further, images in higher resolution better deserve the paper. I truly hope the publisher will be able to maintain that quality because this is a paper in which imaging is of crucial importance and neuroradiology of the skull base is far from obvious for most potential readers. 

Minor changes needed during editing.

Author Response

Dear Reviewer,

thank You for Your suggestions.

urther, images in higher resolution better deserve the paper. I truly hope the publisher will be able to maintain that quality because this is a paper in which imaging is of crucial importance and neuroradiology of the skull base is far from obvious for most potential readers. 

Files has been added to the manuscript and in ZIP versions to keep quality of Figures
